# Rapid fabrication of complex nanostructures using room-temperature ultrasonic nanoimprinting

Junyu Ge[1,7], Bin Ding[2,3,7], Shuai Hou[4,7], Manlin Luo[5], Donguk Nam[5], Hongwei Duan [4✉], Huajian Gao [1,3✉], Yee Cheong Lam [1✉] & Hong Li [1,5,6✉]

Despite its advantages of scalable process and cost-effectiveness, nanoimprinting faces challenges with imprinting hard materials (e.g., crystalline metals) at low/room temperatures, and with fabricating complex nanostructures rapidly (e.g., heterojunctions of metal and oxide). Herein, we report a room temperature ultrasonic nanoimprinting technique (named nanojackhammer) to address these challenges. Nanojackhammer capitalizes on the concentration of ultrasonic energy flow at nanoscale to shape bulk materials into nanostructures. Working at room temperature, nanojackhammer allows rapid fabrication of complex multicompositional nanostructures made of virtually all solid materials regardless of their ductility, hardness, reactivity and melting points. Atomistic simulations reveal a unique alternating dislocation generation and recovery mechanism that significantly reduces the imprinting force under ultrasonic cyclic loading. As a proof-of-concept, a metal-oxide-metal plasmonic nanostructure with built-in nanogap is rapidly fabricated and employed for biosensing. As a fast, scalable, and cost-effective nanotechnology, nanojackhammer will enable various unique applications of complex nanostructures in optoelectronics, biosensing, catalysis and beyond.

[1] School of Mechanical and Aerospace Engineering, Nanyang Technological University, Singapore, Singapore. [2] Institute of Solid Mechanics, Beihang University, Beijing, PR China. [3] Institute of High Performance Computing, A*STAR, Singapore, Singapore. [4] School of Chemical and Biomedical Engineering, Nanyang Technological University, Singapore, Singapore. [5] School of Electrical and Electronic Engineering, Nanyang Technological University, Singapore, Singapore. [6] CINTRA, CNRS/NTU/THALES, UMI 3288, Research Techno Plaza, Singapore, Singapore. [7] These authors contributed equally: Junyu Ge, Bin Ding, Shuai Hou. ✉email: hduan@ntu.edu.sg; huajian.gao@ntu.edu.sg; myclam@ntu.edu.sg; ehongli@ntu.edu.sg

Nanostructures play a crucial role in the era of nanotechnology[1–3]. For instance, gold (Au) and silver (Ag) nanostructures are widely used to enhance the surface electromagnetic field for surface-enhanced Raman spectroscopy (SERS), which are superior sensing platforms for tracing small amounts of molecules down to the single molecule level[4–7]. Copper (Cu) nanostructures are of vital importance to catalysis, showing much higher activity than their bulk counterparts for renewable energy generation or environmental applications[8,9]. Bismuth (Bi) nanostructures are promising candidates of thermoelectric materials due to their highly anisotropic band structure, large Fermi wavelength, etc[10,11]. Rationally designed complex nanostructures can enable novel functionalities, and thus widen their applications. For example, compared to Au nanostructure alone, Au nanostructures with built-in nanogaps can enhance the local electric field by orders of magnitude, and then dramatically increase SERS sensitivity for biosensing[12]. Interface of Cu–zinc oxide heterojunction nanostructure is highly active for $CO_2$ hydrogenation to methanol[13]. Thus, developing a scalable, fast, energy-efficient, and cost-effective technique for making complex "designer" nanostructures with high precision is of crucial importance.

Solution-phase synthesis of nanostructures including chemical reduction[14], seed-mediated growth[15], microwave-assisted growth[16], and microemulsion-based synthesis[17] is scalable. However, its poor uniformity often necessitates a costly and time-consuming purification process as the synthesized nanostructures could comprise nanoparticles of various shapes and nanowires of various lengths due to the diffusion-limited growth process[18]. Moreover, a complex nanostructure is difficult to be synthesized in solution with precision control. Electroplating of metals into nanostructured templates could lead to metal nanowires and even heterojunctions but with limited choice of metals that can be electroplated[19–22]. Advanced lithography followed by lift-off is another widely used method for fabrication of nanostructures directly on substrates; however, the expensive and time-consuming processes involved make it difficult for cost-effective and scalable production of nanostructures[23,24]. Nanoimprinting is a scalable technique, but it is usually limited to soft materials such as polymers[25]. Advanced nanoimprinting of hard materials is possible with the help of high-energy laser that could melt the material; but it is expensive and energy-inefficient[26,27]. Nanostructured bulk metallic glasses could be obtained by nanomoulding at temperatures higher than the corresponding glass transition temperature; however, it only works for amorphous metals but not crystalline metals[28]. In addition, although glass transition temperature is lower than the melting temperature, it is still a relatively high-temperature process[28,29]. Processing at high temperatures and long processing time runs the risk of recrystallization for amorphous metals.

Conventional wisdom of metal forming suggests that it is impossible to make metal nanostructures out of crystalline and polycrystalline metals by deformation, owing to their much larger grain size than the feature size of the nanostructures[29–31]. A direct superplastic nanoimprinting technique, working at about half of the metal melting point, was recently developed to make metallic nanowires out of polycrystalline metals, representing the state-of-the-art nanoimprinting technology for metal nanowires[32]. However, it still suffers from low energy efficiency, high cost, and long processing time. Moreover, it is challenging to make complex nanostructures such as metallic heterojunctions, because metal alloying is inevitable at high temperatures. Therefore, a method for producing complex nanostructures at room temperature is highly desirable but yet to be demonstrated.

Herein, we introduce an ultrasonic nanoimprinting method (named nanojackhammer) to fabricate complex "designer" nanostructures at room temperature in a fast, scalable, energy-efficient, and cost-effective manner. We investigate the working principle of nanojackhammer by complementary high-resolution transmission electron microscopy (HRTEM) and molecular dynamics (MD) simulation. Additionally, we demonstrate quorum sensing (QS), a bacterial cell–cell communication process, with our imprinted complex plasmonic nanostructures comprising an array of vertically aligned metal-oxide-metal nanorods with built-in nanogaps.

## Results and discussion

**Nanojackhammer.** Conventional jackhammer, invented more than 120 years ago, pierces rock with a very low load that can be operated by a single person[33]. The pivotal features of the technology are (1) the frequent strike (about tens of Hz) that reduces the handling force, and (2) the sharp drill bits that exert extremely high local pressure on rock surface in contact with the bits[34]. Similarly, nanojackhammer makes use of high-frequency ultrasonic strike (about 20 kHz) with concentrated energy (sharp features on nanostructured mold) to deform the material for imprinting. Precisely, nanojackhammer capitalizes on the focused ultrasonic power to drive the materials flowing into the nanostructured mold, offering us the flexibility to make nanostructures out of almost all solid materials regardless of their melting points, chemical reactivity, and ductility. Figure 1 illustrates the process flow of ultrasonic nanoimprinting.

The commercially available anodic aluminum oxide (AAO) templates were used as molds in this investigation. One piece of AAO membrane was placed on the substrate, e.g., a 100-μm-thick commercial metal foil (Fig. 1a); then the stack of AAO mold/ metal foil was placed on the stage under the ultrasonic horn. The horn pressed the AAO mold to transfer high-frequency ultrasonic vibration into high-speed strikes on the metal foil surface (Fig. 1b). As such, the surface layer (up to tens-of-micrometers-thick) of the metal foil was shaped into nanostructures, i.e., nanowires/nanorods here due to the circular pores in AAO. Afterward, a demolding process was carried out to obtain the nanostructures on the metal substrate (Fig. 1c, d). If the nanorods are very short (<5 μm), the mold can be detached from the substrate with the help of solid lubricants (e.g., buckminsterfullerene) pre-applied in the AAO mold (see Supplementary Fig. 1; mild sonication can be helpful). Otherwise, the AAO mold was removed by wet etching in sodium hydroxide (NaOH) solution. The whole computer programmed imprinting process was

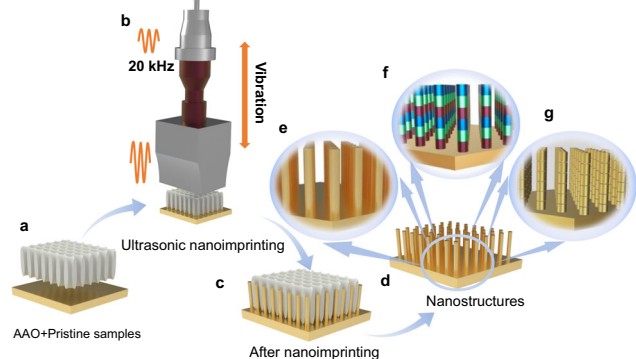

**Fig. 1 Schematic of ultrasonic nanoimprinting of one-dimensional nanostructures. a** AAO mold and metal foil were taken as examples. **b** Ultrasonic nanoimprinting process. **c** Stack of mold and metal foil after nanoimprinting. **d** Metal foil with nanostructures after demolding. **e** Nanowires/nanorods, **f** one-dimensional heterojunctions, and (**g**) nanowires with multiple nanogaps imprinted on substrate surface.

performed and completed within a short duration (from seconds to minutes depending on the sample dimensions).

When the AAO mold/substrate stack was pressed by the ultrasonic horn, the nanoscale wall in the AAO mold served as the "energy director", directing ultrasonic energy flow to the tiny contact area, as illustrated in Supplementary Fig. 2. The sub-100-nm-thick nanowalls in the mold thus acted as "highly energetic nanojackhammers" to deform the metal film by creating dislocations, and then drove it to flow into the nanopores. By controlling the holding force (200–1600 N) and vibration amplitude of the ultrasonic horn (10–30%), the concentrated ultrasonic power through the nanoscale wall was delivered to the substrate to control the imprinting depth (i.e., the length of nanowires/nanorods). The entire process only took several minutes even for 25-μm-long Ag nanowires, as shown in Supplementary Video 1. A sharp contrast is shown in Supplementary Fig. 3, where direct mechanical force alone barely made a dent on the Cu metal foil even when the force was as high as 88,000 N. The AAO broke into pieces when the force was increased to 250,000 N (Supplementary Fig. 3). Thanks to the advances of thin-film deposition technology, our nanojackhammer is able to make "designer" multi-compositional nanostructures through ultrasonic nanoimprinting of a designed multilayer film on the substrate. For instance, pure nanowires/nanorods can be fabricated if a pure metal foil is used as the substrate (Fig. 1e). Multi-segment nanowires consisting of different materials can be obtained on substrate with pre-deposited multilayer thin film (Fig. 1f). Moreover, nanorods with nanogaps can be obtained by depositing sacrificial layers prior to ultrasonic nanoimprinting. After a nanorod with the sacrificial layers is obtained, the sacrificial layers can be partially removed by selective etching, leaving a precisely controlled nanogap between the neighboring layers (Fig. 1g). Free-standing nanostructures can be obtained by selective etching of the sacrificial layer to detach the nanostructures form the substrate. It is worth noting that in principle, the shape of the nanostructure could be varied by using customized mold.

Figure 2 shows various metal nanowires made by our nanojackhammer. Vertically aligned array of Au nanorods (~100 nm in diameter and ~2 μm in length) can be imprinted within 1 min (Fig. 2a). As depicted in Fig. 2b, the grains of the Au nanowires are clearly observed under TEM. In Fig. 2c, the HRTEM images of the nanowire (area enclosed by the red square in Fig. 2b) show perfectly ordered (111) and (200) lattice planes of face-centered cubic Au. Additionally, the length of the nanowires can be tuned by adjusting the process parameters (imprinting force and vibration amplitude) of the ultrasonic imprinting process, as depicted in Supplementary Fig. 4, where the length of Ag nanowires was increased from 200 nm, 2 μm, 5 μm to more than 10 μm. The dependence of the aspect ratio of the imprinted nanowires on the imprinting force is presented in Supplementary Fig. 5. One can observe that, with constant vibration amplitude of 20%, superplastic deformation at room temperature could dominate even when the imprinting force was <700 N. Beyond 700 N, the aspect ratio of the nanowires increased drastically with imprinting force until saturation beyond 1200 N. We then examined the dependence of aspect ratio on the vibration amplitude at constant imprinting forces of 1200 and 1500 N, respectively, as displayed in Supplementary Fig. 6. A linear relationship was found in the range of 10–25% vibration amplitude (recommended safe setting). Therefore, by tuning these parameters, one can control the aspect ratio of fabricated Ag nanowires.

One can vary the diameters of nanowires/nanorods by choosing AAO mold with various pore sizes. For instance, Au nanowires with diameters about 300, 80, 50, and 20 nm (see Supplementary Fig. 7)

were obtained with the corresponding process parameters listed in Supplementary Table 1. In addition, Au nanowires on silicon wafer, tin (Sn), aluminum (Al), Bi, nickel (Ni), and Cu nanowires were imprinted successfully despite the large differences of their physical and chemical properties (Supplementary Figs. 8–13 and Supplementary Table 2), showing excellent generality, versatility, and flexibility of nanojackhammer in making metal nanostructures. More importantly, the fabricated nanowires retain the crystallinity from the starting metal foil, as shown in Supplementary Figs. 14–16.

The other unique feature of nanojackhammer is its capability to fabricate heterojunctions of different materials, even if their mechanical properties are at extreme end of the spectrum. As shown in Fig. 2d, a nanowire array containing Bi–Au heterojunctions was successfully fabricated by depositing Au and Bi layers on a commercial Au foil with Bi and Au being one of the most brittle metals and one of the most malleable metals, respectively. Thanks to the room-temperature ultrasonic nanoimprinting process, abrupt interfaces between Bi and Au was achieved, as shown by the bright field scanning TEM (STEM-BF) and EDS mapping (Fig. 2e, f). Similarly, though there is a large difference between the melting points of Sn (~232 °C) and Au (~1064 °C), abrupt heterojunction of Sn–Au can be made successfully, as depicted in Fig. 2g–i. Other metal heterojunctions including hard-soft metal junction (e.g., Cu–Au), inert-reactive metal junction (e.g., Au–Ag), and metal-plastic junction (e.g., Au-polycarbonate), were also successfully made, as displayed in Supplementary Figs. 17–19, respectively. STEM and EDS mapping show clearly the abrupt interfaces formed at the heterojunctions.

**Atomistic simulations of ultrasonic nanoimprinting**. To rationalize the experimental findings, we conducted large-scale MD simulations to investigate the underlying mechanism of room-temperature nanoimprinting on a single crystal Ag substrate under direct and cyclic loadings. The model setup is illustrated in Supplementary Fig. 20. A dimensionless loading parameter $\delta$ (Eq. (1)) is introduced to characterize the normalized net imprinting displacement during each cycle (Supplementary Fig. 21):

$$\delta = \frac{d_{\text{adv}} - d_{\text{ret}}}{d_{\text{adv}} + d_{\text{ret}}} \tag{1}$$

where $d_{\text{adv}}$ and $d_{\text{ret}}$ denote the imprinting displacements during each loading (advancing) and retreating half-cycles from the Ag substrate. Thus $\delta = 1$ corresponds to "direct loading" without the retreating half-cycles (i.e., $d_{\text{ret}} = 0$), while $0 < \delta < 1$ indicates "cyclic loading" with both loading and retreating half-cycles at a selected ultrasonic frequency appropriate for MD simulations. Figure 3a shows the load-displacement curves at different $\delta$. Compared to direct loading ($\delta = 1$), a significant reduction in the imprinting load has been observed under cyclic loading ($0 < \delta < 1$), in agreement with the experimental observations (Supplementary Figs. 3–5). Figure 3b–i displays the cross-sectional views of local lattice symmetry as well as the volume fractions of dislocated atoms along the length ($z$-axis) of the imprinted Ag nanowire at the end of loading and retreating half-cycles in cycles 5 and 6 under $\delta = 0.14$. Figure 3b, f depicts that, at the end of loading in cycle 5, dislocations are concentrated at the corner (marked by the dashed line) near the root of the nanowire, where severe stress concentration can be expected. The imprint retreats from the Ag substrate by 0.9 nm and some of the generated dislocations at the corner disappear (mainly the 1/6 <112> Shockley partial dislocations), as shown in Fig. 3b, f to c, g. Subsequently, from Fig. 3c, g to d, h, one can see that the imprint changes direction and pushes forward into the Ag substrate by

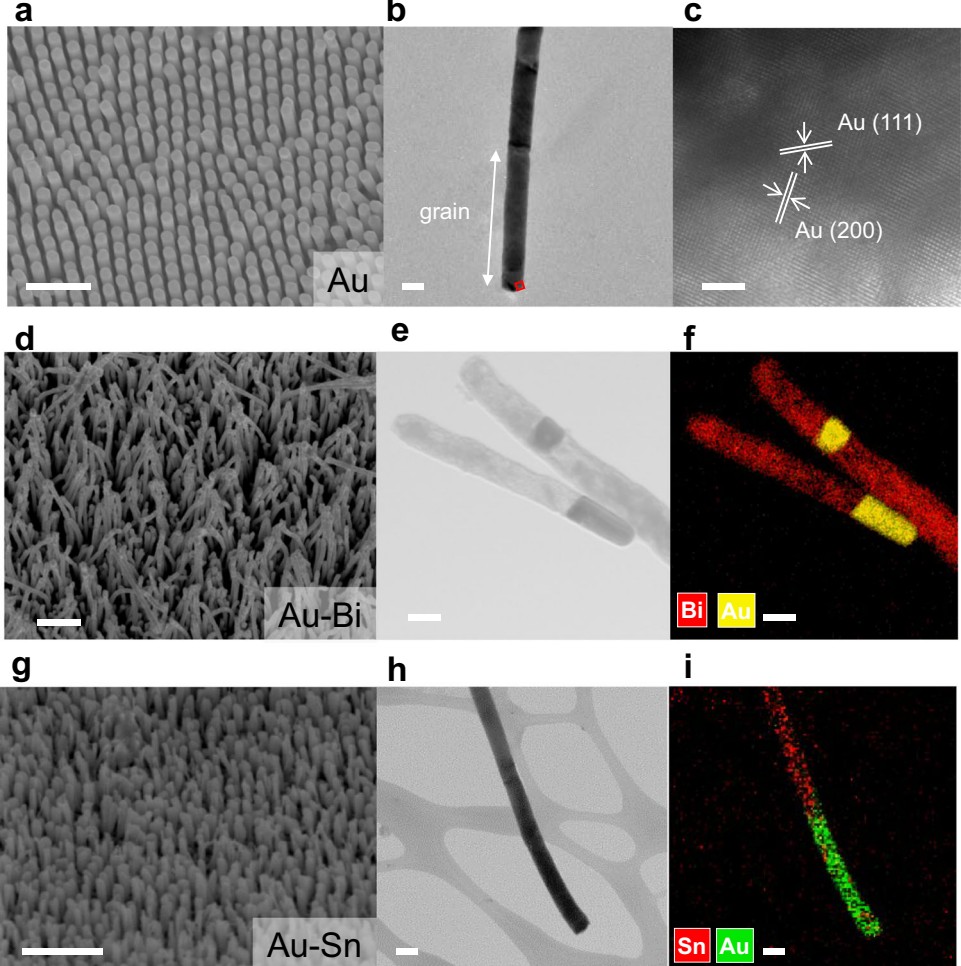

**Fig. 2 SEM and TEM images of metal nanowires. a**, **d**, **g** SEM images of fabricated Au nanowires, Au–Bi, and Au–Sn heterojunction nanowires. **b** TEM image of an Au nanowire. **c** The HRTEM image of the area delineated by the red square in (**b**). **e**, **f**, **h**, **i** STEM and EDS of Au–Bi and Au–Sn heterojunctions. Scale bars, 1 μm (**a**, **d**, **g**), 100 nm (**b**, **e**, **f**, **h**, **i**), and 2 nm (**c**).

1.2 nm, where the residual dislocations at the corner facilitate dislocation generation and then reduce the imprinting load. Afterward, the imprint retreats by 0.9 nm, and some of the dislocations at the corner disappear again, as shown in Fig. 3d, h to e, i. As a result, the alternating dislocations generation and recovery through the loading and retreating half-cycles cooperatively reduce the imprinting load by weakening the deformation resistance of the material. This cyclic loading induced softening, while explored for the first time here for nanoimprinting, is consistent with the acoustoplasticity theory[35] and nanoindentation effect at nanoscale[36]. In the case of direct loading (Supplementary Fig. 22), dislocations continuously nucleate, glide, accumulate, pile-up and interact, leading to much greater resistance to plastic deformation and a significantly higher imprinting load. More importantly, imprinted nanowires are more homogeneous under cyclic loading (Supplementary Fig. 23), where dislocation activities are mainly confined to the corner (see more details about the deformation under $\delta = 1$, 0.14, 0.05 in Supplementary Videos 2–4). We further verified the generality and universality of the observed alternating dislocation generation and recovery mechanism under cyclic loading by considering a mold of a slightly different shape (Supplementary Fig. 24), different mold morphologies (Supplementary Fig. 25), as well as a standard nanoindentation setup, as shown in Supplementary Figs. 26 and 27. Figure 3j summarizes the steady-state mean imprinting force as a function of the loading parameter $\delta$. One

can see that the imprinting force increases monotonously as $\delta$ increases, with direct loading ($\delta = 1$) corresponding to the largest imprinting force. In the latter case, the force often exceeds the machine limit and thus needs to be reduced through heating[32]. However, small $\delta$ indicates slower imprinting speed; thus one should balance the trade-off between imprinting force and speed. Additionally, the imprinted metal heterojunction (Au–Ag) shows the interface was flat and abrupt (Supplementary Fig. 28), consistent with those experimental results shown in Supplementary Fig. 18. It is worth noting that all of these results agree well with experimental observations qualitatively.

**Plasmonic nanostructure for quorum sensing**. As a proof-of-concept, we demonstrate QS sensing using a SERS platform that consists of plasmonic nanostructures made by nanojackhammer, as illustrated in Fig. 4a. QS is a phenomenon of cell-to-cell communication, which is often employed by bacteria to detect and respond to cell population density by producing, secreting, and sensing autoinducer[37–39]. *Pseudomonas aeruginosa* was chosen as a model bacterium to form biofilms on the sensing platform as its QS systems are well established[40,41]. The QS-controlled metabolite pyocyanin is Raman active, and its Raman signals can be measured to reflect the level of QS inside the biofilms. As the sensing platform, an array of vertically aligned Ag nanorods, which contain Ag-alumina-Ag heterojunctions, was

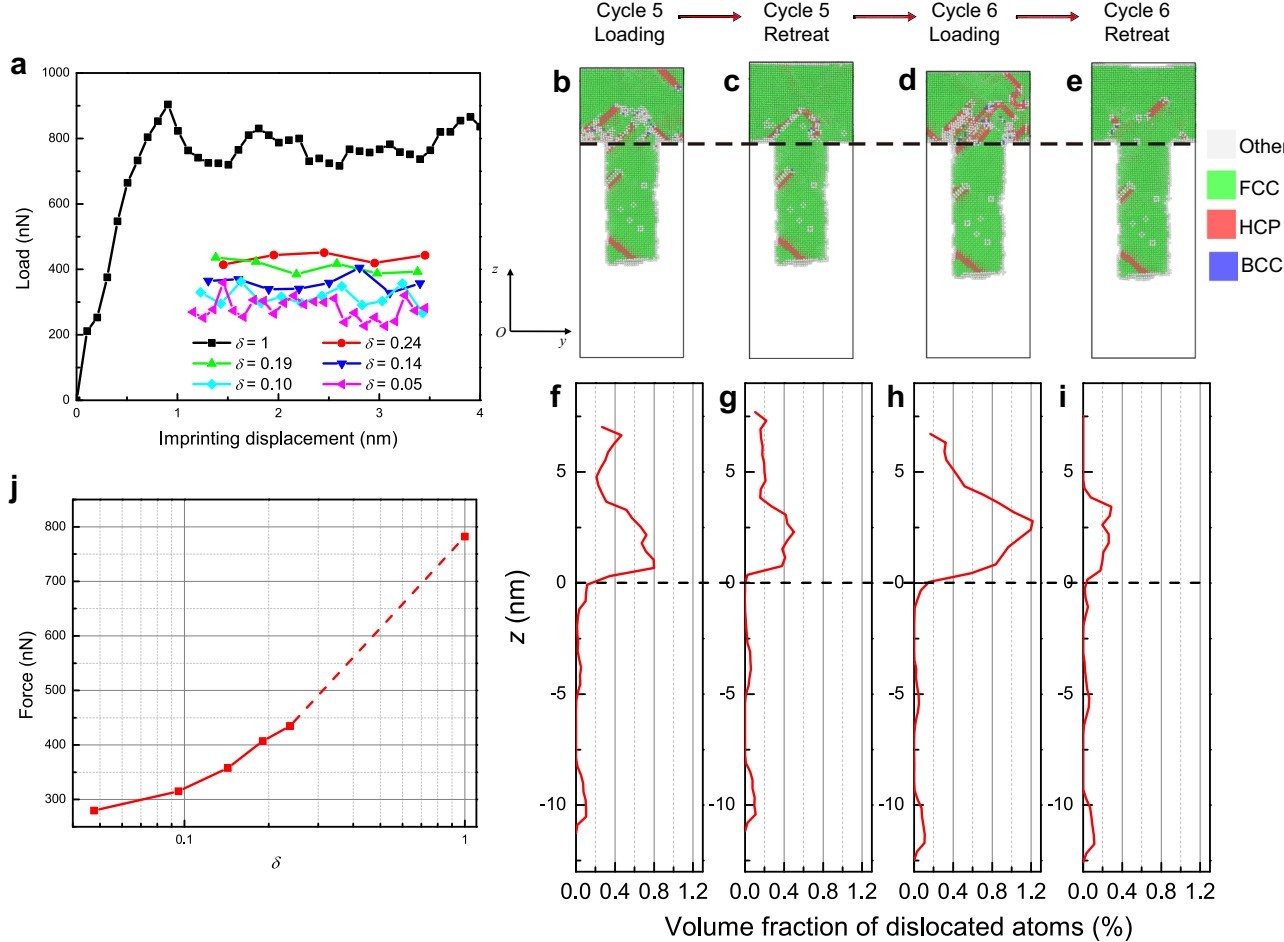

**Fig. 3 Atomistic simulations of nanoimprinting on silver substrate under direct ($\delta = 1$) and cyclic ($0 < \delta < 1$) loading patterns. a** Load-imprinting displacement curves. Under direct loading ($\delta = 1$), the imprinting load is monitored every 0.1 nm of the displacement. Under cyclic loading ($0 < \delta < 1$), the imprinting load is defined as the mean force during each loading half-cycle. **b-e** Sectional views of the deformed silver substrate at imprinting cycles 5 and 6 under $\delta = 0.14$. Atoms are colored according to their local lattice symmetry. **f-i** Volume fraction of dislocated atoms (HCP and BCC type) along the length (z-axis) of the imprinted nanowire at moments corresponding to (**b-e**). The dashed lines in (**f-i**) and in (**b-e**) indicate the position $z = 0$. **j** the steady-state mean imprinting force as a function of the loading parameter.

imprinted from Ag-alumina-Ag trilayer thin film deposited on Ag foil substrate. After a partial etching of alumina layer in NaOH solution, the nanorods (about 360-nm long and 130-nm-thick) with 13-nm-wide nanogaps in the middle were obtained, as depicted in Fig. 4b. Pure Ag nanorods with the same dimensions but without the nanogaps were fabricated for comparison (Supplementary Fig. 29). Numerical calculations revealed that the electric field enhancement inside the nanogap was up to a factor of 1180, as shown in Fig. 4c. In contrast, the electric field of pure Ag nanorods was enhanced by only 15 times between the nanorods (see Supplementary Fig. 30). To experimentally verify that the Raman enhancement was originated from the nanogap, two groups of samples were prepared for comparison: (S1) Raman dye 4-mercaptopyridine (4-MPy) adsorbed on the entire surface of the nanorods (including Ag surfaces and nanogaps), and (S2) 4-MPy selectively adsorbed inside the nanogaps, achieved by blocking the nanorod surface with thiolated poly-ethylene glycol (HS-PEG) before etching alumina (to form the nanogap). The Raman intensity was found to be similar in S1 and S2, suggesting the dominant role of the nanogap in SERS signal enhancement. In contrast, the Raman intensity of 4-MPy was greatly reduced after blocking the nanorod surface with HS-PEG in the control sample of Ag nanorods without nanogaps (see Supplementary Fig. 31).

After culturing bacteria on the sensing platform for 18 h, aggregation of bacteria was evident in the SEM image (Fig. 4d), indicating the formation of a biofilm. The nanorod with built-in nanogaps exhibited a strong Raman spectrum (Fig. 4e), which agreed well with that in a previous report[37], where the sensing platform comprises supercrystals of more than 50 layers of closely packed Au nanorods (about 91-nm-long and 27-nm-thick each) fabricated in a few days. In apparent contrast, our sensing platform was fabricated by nanojackhammer in a few minutes, thus much more facile and cost-effective. Moreover, the plasmonic nanostructure in the platform can be flexibly designed, and then precisely fabricated with high uniformity across a wafer-scale substrate, thanks to the state-of-the-art thin-film deposition technology with atomic-scale precision. The Raman intensity was homogeneous as reflected by the similar spectra collected from six spots across the sensing platform, which showed a variation of <10% (Supplementary Fig. 32). In contrast, pure Ag nanorod substrate showed a greatly attenuated Raman spectrum with poor signal-to-noise ratio, because of the much smaller electric field enhancement. The successful detection of QS-modulated molecules suggests that our method can be further applied to screen potential QS inhibitors for bacterial infection treatment[42,43]. Erythromycin has been demonstrated to inhibit QS in *P. aeruginosa* biofilms at a concentration far below (1/20 of)

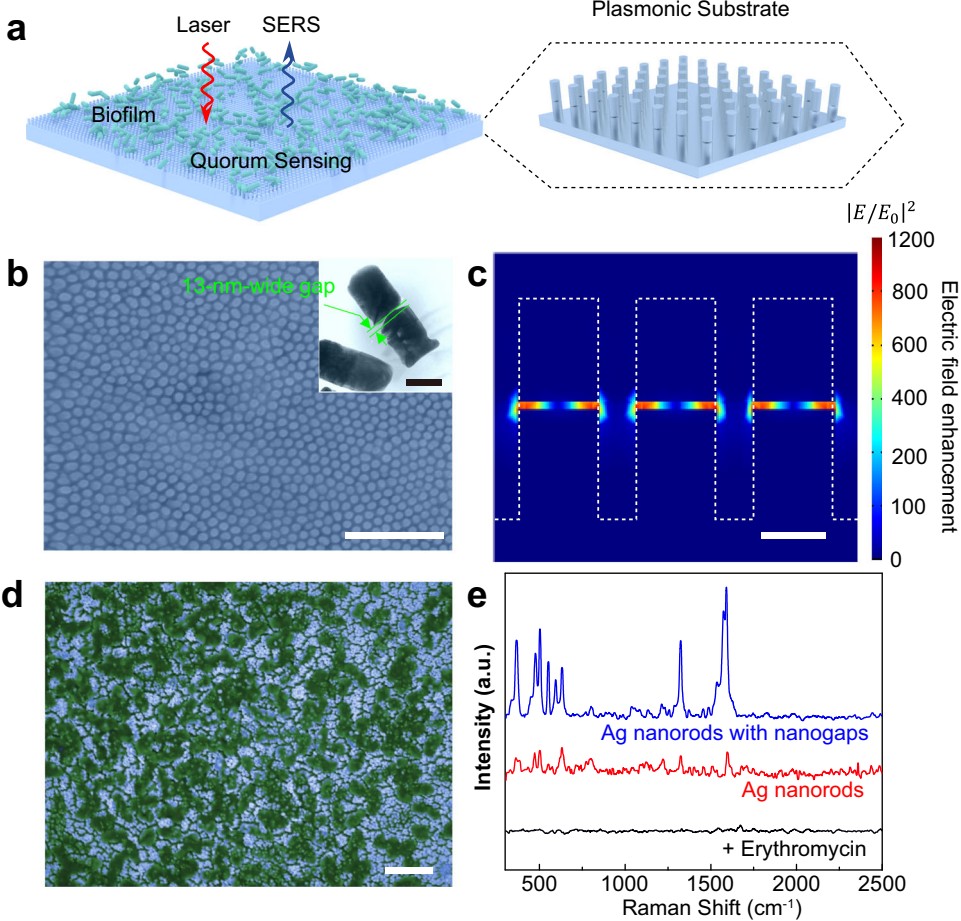

**Fig. 4 Detection of quorum sensing in *P. aeruginosa* biofilms using the fabricated plasmonic nanostructure substrate. a** Schematic illustration of detection of quorum sensing regulated metabolites in biofilms by using surface-enhanced Raman spectroscopy. **b** SEM image of the Ag nanorod array with built-in nanogaps. Inset: TEM image showing a built-in 13-nm-wide gap in the middle of an individual nanorod. **c** Calculated electric field intensity in the nanorod array. Color scale shows the calculated electric fiend enhancement factor. **d** Pseudocolour SEM image of the *P. aeruginosa* biofilm grown on the plasmonic substrate. **e** Raman spectra of the biofilm grown on the gapped Ag nanorod substrate (blue line), Ag nanorod substrate (red line), and the gaped Ag nanorod substrate with 100 μg ml$^{-1}$ erythromycin (black line). Scale bars, 2 μm (**b**, **d**) and 100 nm (**c**, inset of **b**).

the minimal concentration for killing the bacteria[44]. We added 100 μg ml$^{-1}$ erythromycin into the growth media, and the SEM image showed that the bacteria formed a biofilm after 18-h bacterial growth similar to that without erythromycin (Supplementary Fig. 33). However, no Raman signal of pyocyanin was observed (see Fig. 4e), verify the observed Raman signal indeed came from QS sensing.

The advantages of nanojackhammer are summarized as follows. Firstly, the process is carried out at room temperature, largely avoiding oxidation and alloying of reactive materials especially metals. Secondly, the operating conditions of low temperature and atmospheric pressure enable fast, cost-effective, and energy-efficient process. Thirdly, combining with conventional microfabrication technology including atomic layer film deposition and selective etching, complex "designer" nanostructures can be made with nanoscale precision. Last but not least, the process can be scaled up conveniently by automating the imprinting process using large ultrasonic horn and mold. Therefore, nanojackhammer holds great promise to realize unprecedented applications that are impossible with existing or near-term technologies.

In summary, we have demonstrated a room-temperature ultrasonic nanoimprinting technique that works as a jackhammer at nanoscale, where ultrasonic energy is concentrated locally onto the substrate by "energy directors" to shape the substrate surface

into nanostructures. MD simulation reveals the nanojackhammer involves a unique alternating dislocation generation and recovery mechanism. Nanojackhammer offers an ultrafast approach for manufacturing various nanostructures with controlled dimensions and freely chosen materials. Combining with conventional microfabrication processes, this technique allows complex multi-compositional nanostructures to be fabricated with nanoscale precision rapidly (on the time scale of a minute). As a proof-of-concept, reliable QS sensing is demonstrated on an array of vertically aligned Ag-alumina-Ag nanorods comprising built-in nanogaps with excellent uniformity across the sensing platform. Our work opens a route for fast, cost-effective, energy-efficient, and scalable manufacturing of "designer" nanostructures that will enable new applications beyond biosensing.

## Methods

**Materials and characterization.** AAO molds were purchased from JINGYUAN Nanotech Company. Hundred-μm-thick commercial metal foils (Au, Ag, Cu, Sn, Al, Bi, and Ni; purity of 99.999%) were purchased from JIARUN Metal Materials Company. Chemicals (isopropyl alcohol, ethanol, etchant solution) were purchased from Sigma-Aldrich Chemicals Company. Silicon wafers were purchased from UniversityWafer, Inc. Metal pellets (99.999%) were purchased from Lee & Lim International Company. SEM images were investigated by JEOL 7600F instruments. TEM, high-resolution TEM and EDS mapping images were measured by JEOL 2100F with an acceleration voltage of 200 kV. Cross-sectional TEM samples for crystallinity study were cut by Zeiss Crossbeam 540 focused ion beam. Before cutting, a platinum (Pt) layer was deposited to protect the surface of nanowires.

The structural characteristics of single crystal Cu were measured by X-ray diffraction at room temperature on a Paralytical Xpert Pro instrument.

**Ultrasonic nanoimprinting of polycrystalline metal nanowire/nanorods.** All nanowires/nanorods fabrication was carried out by Branson 2000X ultrasonic embossing equipment (20 kHz). AAO molds and metal foils were cleaned subsequently by 18 MΩ cm deionized (DI) water, isopropyl alcohol, and ethanol before use. Before ultrasonic nanoimprinting, metal foils and AAO molds were cut into square chips with a side length of ~5 mm. After ultrasonic nanoimprinting, the AAO molds/substrate stack were immersed in 1 M NaOH solution for up to 2 h to remove AAO and then expose the nanowires. For Al nanowires, a mixture of 1.5 wt% chromic acid and 6 wt% phosphoric acid was used to remove the AAO molds. The stack of Al substrate and AAO molds were immersed in the mixture for up to 3 h at 70 °C.

**Fabrication of nanorods containing heterojunctions.** The Au, Ag, Cu, Bi, Sn metal films were deposited on the substrate at the rate of 1 Å s$^{-1}$ by Edwards Auto 306 E-beam Evaporator before ultrasonic nanoimprinting. Heterojunctions were formed in AAO molds with nanopore diameter ~100 nm. With the holding time setting at 30 s, different heterojunctions were prepared by changing holding forces and amplitudes. To prepare the QS sensing platform, metal layers were deposited by E-beam Evaporator and $Al_2O_3$ layers was deposited by Cambridge Nanotech Atomic layer deposit system at 250 °C at constant growth rate of 1 Å per cycle. AAO molds with nanopore size ~130 nm was used and the holding force was set at 1000 N. After demolding, all samples were cleaned by DI water for several times.

**Atomistic simulations.** To complement the experiments, we performed a series of large-scale atomistic simulations for nanoimprinting and nanoindentation on an Ag substrate by using the large-scale atomic/molecular massively parallel simulator[45]. In all the simulations, the embedded-atom method (EAM) potential[46] was adopted to describe the interatomic interactions in the Ag substrate. This potential can accurately capture the lattice parameter, cohesive energy, elastic constants, phonon frequencies, lattice-defect energies, and energies of alternate structures of Ag, making it suitable for simulating plastic deformation. The Lennard–Jones (LJ) potential was used to describe non-bonded van der Waals interactions between the Ag substrate and the imprinting mold. The parameters used in LJ potentials are listed in Supplementary Table 3. The simulated samples were $10 \times 10 \times 10$ nm$^3$ in size, containing 55,296 Ag atoms randomly oriented with respect to the x-, y- and z-axes along the [100], [010], and [001] directions, respectively. The samples were first relaxed at room temperature (300 K) for 100 ps under an isothermal-isobaric (NPT) ensemble, and then placed on top of a rigid mold that mimics the experimental setup. To mimic the AAO mold used in the experiments, a rigid mold was constructed. For convenience, the mesh of a hexagonal lattice of a single layer graphene with a center hole connected to a carbon nanotube (CNT) was employed. Supplementary Fig. 20b showed a top view of the atomic configuration of the mold. To investigate the effect of mold shape, we also constructed another mold with obtuse corner (102°) by inserting a graphene funnel between the planar graphene and CNT (Supplementary Fig. 24b). Due to the obtuse corner, this mold is expected to induce less stress concentration during the imprinting process. Supplementary Fig. 25 further emphasizes that the physical properties of graphitic carbon are not utilized in the model other than the shape (as a rigid object). A virtual planar indenter (with force constant of 1000 eV Å$^{-3}$) on the top surface of the Ag samples was used to impose the imprinting displacement according to a selected loading pattern (Supplementary Fig. 21). Under loading, the imprint (or indenter) was moved by 0.01 nm every 1000-time steps, corresponding to an average imprinting speed of 10 m s$^{-1}$. The total imprinting displacement was set to 4 nm. The time duration, which clearly depends on δ, is shortest under direct loading (δ = 1). In all the cyclic cases, the cycling frequency was taken as $4.76 \times 10^9$ and the period as 210 ps. Thus, the scalar displacement per cycle $d_{adv} + d_{ret}$ was 2.1 nm, and the net displacement per cycle was controlled by adjusting the difference between the loading and retreating displacements. Note that MD simulations are typically limited to strain rates much higher than those in the experiments but nevertheless have been widely used to capture the essential mechanisms of plastic deformation of many metals which are known to remain qualitatively similar over a broad range of strain rates[47]. Periodic boundary conditions were imposed in the x- and y-directions of the simulation systems. In the z-direction, the boundary condition is non-periodic and shrink-wrapping, instead of being rigid, which means that the position of the face is set as to encompass the atoms in the z-dimension. The temperature was maintained at 300 K via a canonical (NVT) ensemble (see Supplementary Text 1, Supplementary Table 4, and Supplementary Video 5). The integration time step was taken as 1 fs.

To demonstrate the generality of the observed alternating dislocation generation and recovery mechanisms under cyclic loading, we also performed two simulations of standard nanoindentation on Ag, one at 300 K (Supplementary Fig. 26) and the other at 0.1 K (Supplementary Fig. 27). We first constructed Ag substrates with the same size and orientation as those used in the nanoimprinting simulations and relaxed the samples for 100 ps under NPT at 300 and 0.1 K. After equilibration, a virtual rigid spherical indenter with diameter of 6 nm was placed on top of the Ag substrate. In the simulation at 300 K (Supplementary Fig. 26), we

adopted two loading patterns (δ = 1, 0.05) by moving the spherical indenter in the same way as that in the nanoimprinting simulations. In the simulation at 0.1 K (Supplementary Fig. 27), the loading patterns (δ = 1, 0.05) were the same as those used at 300 K with the spherical indenter moved 0.01 nm every 1000-time steps, except every movement of the indenter was followed by energy minimization with energy tolerance setting at 10$^{-4}$ and force tolerance at 10$^{-6}$ eV Å$^{-1}$. Periodic boundary conditions were imposed in the x- and y-directions.

For the formation of Ag–Au heterojunctions under cyclic loading (Supplementary Fig. 28), the substrate was replaced with an Ag film stacked on an Au film, each having thickness of 5 nm. We used EAM potential[48] and LJ potentials (Supplementary Table 3) to describe the interatomic interactions in the Au film and non-bonded van der Waals interactions between different materials (the Ag film, the Au film, and the mold). All other settings were kept the same as the nanoimprinting simulations of a pure Ag film. We did not simulate the heterojunction of Bi–Au and Sn–Au due to the lack of accurate MD potentials for Bi and Sn.

**Simulation of electric field on nanorod array substrates.** Finite element method simulation was performed to determine the electric enhancement in the nanorod arrays in the RF module of COMSOL Multiphysics 5.5. The nanorod array supported on a silver substrate was assumed to be in a hexagonal packing arrangement with a surface-to-surface separation of 50 nm. The nanorod was modeled as a silver cylinder (130 nm in diameter and 360 nm in height) with an alumina layer (13 nm in thickness) exactly in the middle along the height. A notch with a depth of 20 nm was created in the alumina layer to represent the etched nanogap. The relative permittivity of alumina and silver were set as 3.10 and −29.96–0.38$i$, respectively. A cuboidal unit cell of the nanorod array was created, and all its four side surfaces perpendicular to the substrate was applied with a periodic boundary condition. A plane wave light source with a wavelength of 784.5 nm was applied from a port above the nanorod array with the incident direction perpendicular to the substrate and the electric field polarization parallel to the side of the unit cell. After dividing the system into tetragonal meshes, a MUMPS direct solver was used to solve the Maxwell equations to obtain the electric field on each node of the system.

**Biofilm growth and Raman-based detection of QS.** PAO1, a common *P. aeruginosa* strain, was cultured on the plasmonic substrates by adding 10$^8$ colony-forming unit of mid-log phase bacteria in 20 μl of Luria–Bertani (LB) broth and incubated at 37 ºC in a humid environment for 18 h. To explore the effect of erythromycin on the modulation of QS, 100 μg ml$^{-1}$ erythromycin was added in LB broth. The Raman spectra of the biofilm were captured on a Raman microscope (Supplementary Figs. 32–34) with a 784.5 nm laser (120 mW) and exposure time of 10 s.

## Data availability
The data that support the findings of this study are available from the corresponding authors on request.

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

## Acknowledgements

This work was supported by Nanyang Technological University under NAP award (M408050000), and Singapore Ministry of Education Tier 1 program (2018-T1-001-051). The authors acknowledge the Facility for Analysis, Characterization, Testing and Simulation (FACTS), Nanyang Technological University Singapore for use of electron microscopy and X-ray facilities. B.D. and H.G. are grateful for a research start-up grant (002479-00001) from Nanyang Technological University and the Agency for Science, Technology and Research (A*STAR) and the use of the A*STAR Computational Resource Centre, Singapore (ACRC) and National Supercomputing Centre, Singapore (NSCC).

## Author contributions

J.G., Y.C.L., and H.L. conceived the idea and designed the experiment of ultrasonic nanoimprinting. J.G. performed the ultrasonic nanoimprinting and material characterizations. B.D. and H.G. conducted the MD simulation. S.H., J.G., and H.D. designed and conducted the QS sensing experiment. M.L., D.N., and J.G. carried out the nanogap sample fabrication. J.G., B.D., S.H., and H.L. wrote the manuscript, and all authors discussed the results and commented on the manuscript.

## Competing interests

J.G., H.L., and Y.C.L. are inventors on a patent application related to this work, filed by Nanyang Technological University Singapore (application no. 10202008172X). The authors declare no other competing interests.
