## [Peer Review File · Nature Communications]

REVIEWER COMMENTS

Reviewer #1 (Remarks to the Author):

This manuscript describes an ultrasonic nanoimprinting technique to fabricate metal nanostructures at room temperature. Though similar processes have been reported in literatures, and ultrasonic fascinated forming has also been well developed, direct applying nanoimprinting to metals at room temperature (without the aid of ultrasonic) is general challenging. This work reports some useful discoveries that are of interest to the scientific community. The work can be considered for publication after some major revisions.

Major:

1. The authors rationalized the successful R.T. nanoimprinting as the mold cavity directed ultrasonic energy flow. Alternatively, the reviewer speculates that plastic deformation induced heating may also contribute to the metal flow since the R.T. nanoimprinting is essentially a severe local plastic deformation process.
2. It has been well known that ultrasonic vibration assisted forming can enhance the deformability of materials. The enhanced deformability has been attributed to vibration induced stress superposition, acoustic softening or friction decrease. In the MD simulations, the authors used graphene/carbon nanotube as mold. Why not use the mold material in experiments (i.e. Al₂O₃)? It is known that carbon materials are good lubricants, this can affect the interface friction.
3. Again, the reviewer considers the loading induced local heating effect in the simulations. The effect of loading rate and monitoring the sample temperature distribution (especially at the corners) should be provided.
4. Based on the MD simulations, the authors observed and concluded that the residual dislocations after the retreating half cycles at the corner facilitate dislocation generation and then reduce the imprinting load. This is reasonable by comparing the amount of dislocations in Fig. 3f, g and h. However, it seems that the amount of residual dislocations after the cycle6 unloading (Fig. 3i) is much less than those after the cycle5 unloading (Fig. 3g). The authors are suggested to compare dislocations within more of cycles such as cycle7. The following question is why the authors selected to show dislocations at the cycles of 5 and 6?

Minor:

1. The metals described in this paper are all soft metals. It has been recently shown that Au can be nanoimprinted at -60 °C without vibration (PRL, 122, 016101, 2019). Can hard materials such as nickel or iron be prepared at R.T. by the proposed nanojackhammer method?
2. The authors used chromic acid to demoulding Al nanowires. Why the chromic acid doesn't corrode Al (an amphoteric metal)? What's the concentration of the used chromic acid? The authors should give experimental details.
3. At pages 5-6, the authors claimed that the mold could be recycled since it can be detached from the substrate by itself with the help of solid lubricants. The reviewer suggests to delete it since the used thin AAO film is really brittle, the imprinting or demoulding processes can damage it. Besides, the AAO mold

can also be contaminated by the solid lubricants. Otherwise, the following question is how many times could the AAO film be reused?

4. At page 8, the authors claimed that “the fabricated nanowires retain the crystallinity from the starting metal foil, as shown in Supplementary Figs. 12 and 13.” Does it mean the crystalline orientation of the nanowire follows that of the contact metal surface? It is clear that the Supplementary Figs. 12 and 13 can’t support this conclusion. One possible strategy to check this is cutting a nanowire attached in a thin substrate by FIB and follows to characterize both the substrate and the nanowire under TEM.

5. The reviewer noted that only imprinting force was given (e.g. page 7) in the experiments, this doesn’t make sense if it is not related with the sample size.

6. Nanoimprinting of pre-deposited multilayer thin film may introduce uniformity issues because of the rough surface of sample, mold, or indenter.

7. The authors are suggested to give the force constant for the indenter surface in the MD methods.

8. The authors show the averaged imprinting force (mean force during each loading half-cycle) in the simulations (e.g. Fig. 3j). Can the authors show the time dependent reaction force within one cycle?

Reviewer #2 (Remarks to the Author):

Ge et al. report a novel ultrasonic nanoimprinting method that produces nanostructures from all solid materials despite their chemical reactivity. In brief the technique consists in placing an AAO template in contact with the foil to be patterned and apply an ultrasonic horn. The process shows high resolution (replicating the AAO template) with high aspect ratios (up to 100 microns length nanowires). The “nanojackhammer” can also produce nanowires from multilayered foils as demonstrated with some examples. The authors also report the physical mechanisms underlying the technique and fabricate and test a plasmonic structure for the SERS detection of pyocyanin.

In general, this is an interesting novel nanofabrication technique. Its main advantages are that it allows rapid templating with nanoscale features of a given solid at RT and within minutes. However, it also faces some drawbacks such as demolding requires in some cases the dissolution of the template. The authors have shown a variety of nanostructures possible, these nanowires can also be obtained with alternative methods (eBL + RIE), although with additional cost and time. In sum, this technology can be interesting for the community, this is why this manuscript may become acceptable for publication should the following comments be addressed by the authors:

Q1. An interesting advantage of the technique is that it preserves the crystal structure of the original foil, however I do not see that this benefit has been exploited fully by the authors to produce single crystal structures. Could single crystal nanostructures be achieved if used a single crystal as foil, such as a silicon wafer or an epitaxial gold film?

Q2. What is the interplay between the hardness/mechanical properties of the alumina template and the target material? could other materials work as stamps or this technique works only with AAO as stamp? AAO is limited to hexagonal arrays and the domains are rotated to each other... Is it possible to replicate EBL defined motifs?

Q3. Is there conformal contact between AAO molds and metal foil achieved during the printing process?

Is it required as in other nanoimprinting methods?

Q4. The authors mention a Au-polycarbonate nanowire array. Could the authors elaborate more on the application of their method to organic materials?

Q5. The authors mention in their ref 43 another system to measure QS. How does their structure compare in terms of sensing performance to this reference?

Comment. The manuscript is well written and organized. However, in some parts, I would appreciate a bit more experimental detail, i.e.: in page 5, the authors describe the nanoimprinting process of a metal foil, but the metal and how was it obtained are not specified therein. Similarly, on page 8, the nanostructuring process of an Au-Bi multilayer lacks the information of this material. All this info can be found at the experimental section, but a bit more information helps the reader to grasp the idea more rapidly.

Reviewer #3 (Remarks to the Author):

In this manuscript, the authors propose a scalable and rapid nanostructures-imprinting technique, named nanojackhammer. Using ultrasonic cyclic loading, the authors demonstrate that the imprinting technique works well even at room temperature for several metals (Ag, Au, Cu, Sn, Bi), metal-metal-oxide, and metal-plastic and fabricates length-controlled nanowires having complex nanostructures. Besides, the underlying severe plastic deformation mechanism is discussed based on atomistic behavior in the substrate obtained by molecular dynamics imprinting simulations for FCC metals. This is well-written manuscript with novel technique and demonstration that could become publishable in Nature Communications. However, before that, I suggest the following points need to be addressed to strengthen the paper's discussion.

1. For the cyclic loading frequency, the authors applied an ultrasonic frequency of 20kHz. Is this the best frequency for any imprinting materials, sizes (mold shape), and loading amplitudes? I suggest a frequency-dependent imprinting efficiency can be presented and discussed, which should be valuable information in this technique's practical use. Moreover, since these factors determine strain rate distribution generating in the imprinting material, the information may also strengthen the mechanism discussion based on the MD simulations with a much higher GHz frequency.

2. The authors performed MD simulation on FCC metals and discuss the imprinting mechanism based on dislocation behavior during the cyclic loading, such as alternating dislocation generation and recovery mechanism. As the authors mention in the manuscript, even a significantly higher strain rate was used in MD simulation. I believe that the simulations should demonstrate the essence of the imprinting mechanism of FCC metals and maybe glimpse the fundamental physics of the mean force reduction. However, imprinting not only FCC metals but also oxide and plastic materials, and perhaps also ceramics and glasses, should be one of the advantages of this imprinting technique. Since the non-FCC solid materials generally have different plastic deformation mechanisms in such nano-scaled confined volume, more general discussion, which is applicable even for the non-FCC materials, is needed.

3. Minor points:

(i) In the MD simulation, the effect of boundary conditions (BC) applied to the substrate (PBC in x and y and maybe almost rigid in z) should be carefully checked. When the imprinting proceeds, if the total number of atoms is constant as in this simulation, the substrate thickness must be reduced, and the BC effects should be changed accordingly.

(ii) For Fig.3, the definition of volume fraction of dislocation is not clear.

Point-by-Point Response to the Reviewers' Comments (NCOMMS-20-34796)

We thank all the reviewers for their insightful comments and constructive suggestions, which have allowed us to further improve our manuscript. We have carefully revised our manuscript according to their comments. Our point-by-point responses are as follows.

REVIEWER COMMENTS

Reviewer #1 (Remarks to the Author):

This manuscript describes an ultrasonic nanoimprinting technique to fabricate metal nanostructures at room temperature. Though similar processes have been reported in literatures, and ultrasonic fascinated forming has also been well developed, direct applying nanoimprinting to metals at room temperature (without the aid of ultrasonic) is general challenging. This work reports some useful discoveries that are of interest to the scientific community. The work can be considered for publication after some major revisions.

Major:

1. The authors rationalized the successful R.T. nanoimprinting as the mold cavity directed ultrasonic energy flow. Alternatively, the reviewer speculates that plastic deformation induced heating may also contribute to the metal flow since the R.T. nanoimprinting is essentially a severe local plastic deformation process.

Our response:

We thank the reviewer for this important comment. First, we fully agree that R.T. nanoimprinting is essentially a severe local plastic deformation process. For the imprinted metal sample, in our scenario with the imprinting frequency of ~20 kHz, the absorbed energy from sound waves should be negligible [Langenecker, B., *Effects of Ultrasound on Deformation Characteristics of Metals. IEEE Transactions on Sonics and Ultrasonics 1966, 13 (1), 1-8.*]; therefore, the energy change ΔE consists of the work done by imprinting W and the released heat H , *i.e.*, $\Delta E=W+H$. The released heat H mainly comes from the local plastic energy dissipation at the corner.

To address the question on the role of H , we have performed new theoretical study, as shown in revised Supplementary Information. Our new theoretical analysis on the thermal transport during ultrasonic imprinting in **Supplementary Text 1** suggests that heat rise in the metal sample has little effect on the deformation behavior and the underlying mechanism. Rather, it shows that in an open environment, local heat would transport away quickly enough (much shorter than one cycle of 0.00005 s), and the metal sample would stay near the ambient temperature (room temperature) throughout the imprinting process. Moreover, our theoretical analysis is consistent with the result from a similar experiment that has monitored the temperature of aluminum sample by an embedded thermocouple in the ultrasonic loading tests. [Daud, Y.; Lucas, M.; Huang, Z., *Modelling the effects of superimposed ultrasonic vibrations on tension and compression tests of aluminum. J. Mater. Process. Technol. 2007, 186 (1), 179-190.*]

Supplementary Text 1. Theoretical analysis of thermal transport in the metal sample during ultrasonic imprinting

During ultrasonic imprinting in an open environment, the associated released heat mainly comes from local plastic energy dissipation at the corner. Taking Ag sample as an example, the local plastic energy induced heat transport in the NW-axial direction can be analyzed.

Schematic of model set up

In the metal sample, thermal transport along the axial direction can be simplified as

$$\frac{\partial u}{\partial t} = \alpha \frac{\partial^2 u}{\partial x^2}, \alpha = \frac{k}{\rho C_p} \quad (1)$$

where u , t and x are the temperature, time and coordinate, respectively. Parameters α , k , ρ , C_p are used to denote the thermal diffusivity, thermal conductivity, density, and specific heat capacity, respectively. Suppose that after each cycle, all the dislocation energy W_d at the corner ($x=L$) can be harvested to contribute to the local temperature rise Δu , $W_d V = \rho V C_p \Delta u$. We can estimate the local temperature rise at the corner as

$$\Delta u = \frac{W_d}{\rho C_p} \quad (2)$$

where the dislocation energy W_d is the sum of the energy by all dislocations $W_d = \rho_d w_d$. Here, ρ_d is the number of dislocations per unit volume and w_d is the energy of a single dislocation, which includes the elastic strain energy and dislocation core energy.

$$w_d = w_{el} + w_{core} = \frac{Gb^2 l}{4\pi K} \left(\ln \frac{R}{r_0} + Z \right) \quad (3)$$

where G is the shear modulus of the metal sample, b and l are the Burgers vector and the length of the dislocation respectively; $K = 1$ for screw and $K = 1 - \nu$ for edge dislocations. The two items in the bracket correspond to contributions from the long-range elastic field [$\sim \ln(R/r_0) \approx 5-17$] and the dislocation core ($\sim Z \approx 1-3$), respectively. To estimate the maximum-possible temperature rise, we adopted the upper bound values of all parameters $\rho_d = 10^{16} \text{ m}^{-2}$, $w_d = 6 \text{ eV}$ with the Burgers vector $b = 3 \times 10^{-10} \text{ m}$, and obtained an estimate of the maximum temperature rise $\Delta u \approx 15 \text{ K}$. Then we set the boundary conditions of Eq. (1) as

$$\begin{aligned} u(x, 0) &= \Delta u \sin\left(\frac{\pi}{2L} x\right) + R_0 \\ u(0, t) &\equiv R_0, \quad \frac{\partial u}{\partial x}(L, t) = 0 \end{aligned} \quad (4)$$

where R_0 is the environmental temperature (room temperature 300 K). For an Ag NW with the maximum length of $\sim 0.045 \text{ mm}$, the numerical solutions below showed that the local heat generated in one cycle is transported away quickly and has negligible effect on the next cycle.

Numerical solution to thermal conduction in the NW-axial direction

Our theoretical analysis is consistent with the result from a similar experiment that has monitored the temperature of aluminum sample by an embedded thermocouple in the ultrasonic loading tests [Daud, Y.; Lucas, M.; Huang, Z., *Modelling the effects of superimposed ultrasonic vibrations on tension and compression tests of aluminum. J. Mater. Process. Technol.* 2007, 186 (1), 179-190.]. Considering the similar thermal properties of common metals as listed in Supplementary Table 4, this theoretical analysis should be applicable to a wide range of metal samples.

Our revision #1: This new modeling results and analysis have been added to revised Supplementary Information (highlight in page 36). The discussion about possible temperature rise has been added to revised Manuscript (highlight in page 17).

2. It has been well known that ultrasonic vibration assisted forming can enhance the deformability of materials. The enhanced deformability has been attributed to vibration induced stress superposition, acoustic softening or friction decrease. In the MD

simulations, the authors used graphene/carbon nanotube as mold. Why not use the mold material in experiments (i.e. Al₂O₃)? It is known that carbon materials are good lubricants, this can affect the interface friction.

Our response:

We thank the reviewer for this valuable comment. Though we call our mold graphene/carbon nanotube, we only used the atomic mesh (for the sake of saving the model setting up time) instead of the physical properties of graphitic carbon. In other word, it is a rigid confinement with hexagonal atomic mesh. Our detailed explanation is as follows.

Since the AAO mold is very hard and stiff, during ultrasonic nanoimprinting, the mold effectively plays the role of deformation confinement to the metal sample and its self-deformation is negligible. To focus on the deformation behavior of the imprinted sample in the MD simulation, it is a well-regarded approximation to assume the mold as a rigid object [Pei, Q., Lu, C., Liu, Z. & Lam, K. *Molecular dynamics study on the nanoimprint of copper. Journal of Physics D: Applied Physics* 40, 4928 (2007); Wu, C.-D., Fang, T.-H., Chiang, C.-C. & Kuo, L.-M. *Effect of mold geometry on nanoformed aluminum films investigated using molecular dynamics simulations. Computational materials science* 74, 17-22 (2013); Wu, C.-D. & Hou, C.-J. *Molecular dynamics analysis of plastic deformation and mechanics of imprinted metallic glass films. Computational Materials Science* 144, 248-255 (2018)]. Under such a rigid condition, the shape of the mold is of utmost importance instead of the material of the mold. In our manuscript, to save simulation time, simple hexagonal atomic mesh associated with

graphitic carbon is employed to mimic the shape of the AAO mold. Such a treatment greatly saves the simulation time without compromising the ability to confine deformation, which is crucial for the underlying working principle.

Indeed, as pointed out by the reviewer, the lubricant can affect the interface friction, as shown in our experiments with buckminsterfullerene. Nevertheless, as mentioned above, the physical properties of graphitic carbon are not utilized in the model rather than the shape (as a rigid object). Moreover, we have conducted additional two simulations in **new Supplementary Fig. 25** to show that, despite the interfacial friction may affect the dislocations morphology in the nanowire, the alternating dislocation generation and recovery behavior at the corner remain essentially the same.

Supplementary Fig. 25 | Comparison of different mold morphologies. Sectional views of the deformed Ag substrate confined by a mold with surface wavelength of (a) 0.1 nm and (b) 1 nm at imprinting cycles 8 and 9 under $\delta = 0.05$. Similar dislocation generation and recovery mechanism is observed in both cases.

Our revision #2: The new simulation results have been added to the revised Supplementary Information (highlight in page 26). The statements emphasizing the shape (instead of physical properties) of graphitic carbon model have been added to revised Manuscript (highlight in page 17).

3. Again, the reviewer considers the loading induced local heating effect in the simulations. The effect of loading rate and monitoring the sample temperature distribution (especially at the corners) should be provided.

Our response:

We thank the reviewer for this constructive suggestion. Monitoring the local temperature is extremely challenging experimentally due to the setup where stack of thin foil and AAO membrane is sandwiched between 3×3 inch² Ti horn and sample chuck. As per reviewer's suggestion, we have tried our best to perform the temperature measurement as follows.

To monitor the local temperature as accurately as possible, we stuck a very thin and sensitive thermocouple on the metal foil. **Supplementary video 5** shows the temperature change under ultrasonic nanoimprinting (amplitude: 25%; force: 1200 N).

The temperature rose from 28.6 to 41.1 °C during imprinting and then rapidly decreases after nanoimprinting, and thus no significant temperature change was observed.

Additionally, based on our theoretical analysis in response to Q1, local plastic deformation induced heating should only cause small rises in temperature due to the rapid heat transport in an open environment. Therefore, the sample temperature distribution does not appear to be a significant contributing parameter for both the experimental and modelling studies. Moreover, in MD simulations, considering different sample temperatures and different loading rates, we have verified the generality of the alternating dislocation generation and recovery mechanism under cyclic loading by nanoindentations (Supplementary Figs. 26 and 27). These include a consideration of two loading patterns ($\delta = 1, 0.05$) at 300 K (Supplementary Figure 26) and two different loading patterns ($\delta \approx 1, 0.05$) at 0.1 K (Supplementary Figure 27); the loading patterns include a spherical indenter moving 0.01 nm every 1000 steps, and additionally each step movement of the indenter was followed by energy minimization with energy tolerance setting at 10^{-4} and force tolerance at 10^{-6} eV/Å.

Our revision #3: The new Supplementary Video 5 with associated discussion have been added to Supplementary Information (page 43).

4. Based on the MD simulations, the authors observed and concluded that the residual dislocations after the retreating half cycles at the corner facilitate dislocation generation and then reduce the imprinting load. This is reasonable by comparing the amount of dislocations in Fig. 3f, g and h. However, it seems that the amount of residual

dislocations after the cycle6 unloading (Fig. 3i) is much less than those after the cycle5 unloading (Fig. 3g). The authors are suggested to compare dislocations within more of cycles such as cycle7. The following question is why the authors selected to show dislocations at the cycles of 5 and 6?

Our response:

We thank the reviewer for careful reading and this important comment. The following figure plots the dislocation evolutions in the following Cycles 7 and 8.

Moreover, Supplementary Videos 2-4 describe the whole imprinting process under the loading pattern $\delta=1, 0.14, 0.05$, respectively, for better visualization of the mechanism. One can observe the dislocation motions not merely at Cycles 5 and 6, but at any two adjacent cycles.

Minor:

1. The metals described in this paper are all soft metals. It has been recently shown that Au can be nanoimprinted at -60 °C without vibration (PRL, 122, 016101, 2019). Can hard materials such as nickel or iron be prepared at R.T. by the proposed nanojackhammer method?

Our response:

In paper (PRL, 122, 016101, 2019), Au nanorods grow through continuous dislocations nucleation and propagation. The process is quite similar to our MD simulation of direct loading ($\delta = 1$) (Supplementary Figure 22). Compared to direct loading, a significant reduction in the imprinting load has been observed under cyclic loading due to the unique alternating dislocation generation and recovery mechanism. Experimentally, hard metals imprinting is more challenging as these metals can absorb more energy from sound waves. [*Langenecker, B., Effects of Ultrasound on Deformation Characteristics of Metals. IEEE Transactions on Sonics and Ultrasonics 1966, 13 (1), 1-8.*]

To address reviewer's question, we have added new results on ultrasonic imprinting of Ni nanowires and long Cu nanowires, as shown in **new Supplementary Figure 12 and Supplementary Figure 13.**

Supplementary Fig. 12 | SEM image and EDS spectra of imprinted Ni nanowires.

a, morphology of Ni nanowires. **b**, EDS spectra of the full area in (a). **c**, the summary table the element distribution. Scale bar, 2 μm (a).

Supplementary Fig. 13 | SEM image of imprinted long Cu nanowires under 1,200

N and 25% amplitude. Scale bar, 1 μm .

Our revision #4: We have added these new results in revised Supplementary Information (Supplementary Figs. 12 and 13). We have also amended the discussion in revised Manuscript (highlight in page 8).

2. The authors used chromic acid to demoulding Al nanowires. Why the chromic acid doesn't corrode Al (an amphoteric metal)? What's the concentration of the used chromic acid? The authors should give experimental details.

Our response:

We thank the reviewer for the constructive suggestion. It is an established selective etching process for anodic aluminum oxide. [Chen, S.H., Wang, C.Y., Chen, Y.C., Hun, C.W., Chen, S.F. and Yang, S.M., (2015). *Fabrication of pure aluminum nanowires by using injection molding process in ambient air. Materials Letters, 148, pp.30-33.*] We employed a mixture of chromic acid and phosphoric acid. Chromic acid is a strong oxidizing agent, which protects the surface of Al nanowires by forming alumina layer while phosphoric acid etches the AAO. During the etching process, the surface of Al nanowires will be oxidized by chromic acid first, then etched by phosphoric acid. Delayed by the oxidation process by chromic acid, the etching rate of Al becomes slower than AAO that experiences direct phosphoric acid etching. Moreover, mass transport of etchant molecules around Al nanowires is very slow due to the limited space around Al nanowires. Therefore, by controlling the etching time, AAO can be removed to obtain Al nanowires.

Our revision #5: We have added the following information to Method section of revised Manuscript (highlight in page 15).

“For Al nanowires, a mixture of 1.5 wt.% chromic acid and 6 wt.% phosphoric acid was used to remove the AAO molds. The stack of Al substrate and AAO molds were immersed in the mixture for up to 3 h at 70 °C.”

3. At pages 5-6, the authors claimed that the mold could be recycled since it can be detached from the substrate by itself with the help of solid lubricants. The reviewer suggests to delete it since the used thin AAO film is really brittle, the imprinting or demoulding processes can damage it. Besides, the AAO mold can also be contaminated by the solid lubricants. Otherwise, the following question is how many times could the AAO film be reused?

Our response:

We thank the reviewer for this constructive suggestion. As per reviewer's suggestion, we have deleted the statements in the revised Manuscript.

Our revision #6:

Supplementary Fig. 1 has been revised to be consistent with the revised manuscript.

Revised Supplementary Fig. 1 | Working principle of ultrasonic nanoimprinting with demolding process. a, Lubricants dropped on AAO mold. b, Ultrasonic

nanoimprinting process. **c**, Stock of mold and metal foil. **d**, Ultrasonic assisted AAO mold and nanowires separation process. **e**, Nanowires.

4. At page 8, the authors claimed that “the fabricated nanowires retain the crystallinity from the starting metal foil, as shown in Supplementary Figs. 12 and 13.” Does it mean the crystalline orientation of the nanowire follows that of the contact metal surface? It is clear that the Supplementary Figs. 12 and 13 can’t support this conclusion. One possible strategy to check this is cutting a nanowire attached in a thin substrate by FIB and follows to characterize both the substrate and the nanowire under TEM.

Our response:

We thank the reviewer for this important comment. The polycrystallinity of imprinted nanowire will be similar to that of the starting foil; however, grain boundary rotation and grain elongation likely occur during imprinting; thus, the exact lattice direction could change after imprinting. As per reviewer’s suggestions, we used FIB to cut a piece of substrate attached with Ag nanowires, and the new results are presented as follows.

New Supplementary Fig. 14 shows the TEM image of bulk Ag foil and attached Ag nanowires. The inserted fast Fourier transformation (FFT) of Supplementary Fig. 14b exhibits typical polycrystalline spots. Supplementary Fig. 14c-e presents HRTEM images of the selected area in Ag nanowire (Supplementary Fig. 14a), and Supplementary Fig. 14f-k depicts the FFT images of Supplementary Fig. 14c-e. The spots of FFT images shows that the fabricated nanowires retain the crystallinity from the starting metal foil.

Supplementary Fig. 14 | TEM and HRTEM images of polycrystal Ag nanowire. a, Ag nanowires rooted on Ag foil fabricated by focused ion beam (FIB). **b,** Topography of the bulk Ag foil, the red arrow in (a) points to the area of (b). The inserted fast Fourier transformation (FFT) image shows the polycrystal nature of Ag foil. **c-e,** TEM images at the regions denoted by A, B, and C in (a). **f-k,** HRTEM images of selected area in (c-e), the spots of the inserted FFT images are different from each other, indicated the polycrystal nature of the Ag nanowire, which proved the fabricated nanowires retain the crystallinity of bulk materials. Scale bars, 20 nm (**a,b**), 10 nm (**c-e**), and 2 nm (**f-k**).

Moreover, we imprinted Cu nanowires on single crystal (111) Cu foil. The X-ray diffraction spectra of the Cu foil (**new Supplementary Figure 15b**) shows only one peak [Cu (111)], indicating the single crystal nature of the Cu foil.

Supplementary Fig. 15 | X-ray diffraction spectra of (a) polycrystal Cu foil and (b) single crystal Cu foil.

After imprinting Cu nanowires from the single crystal Cu foil, we used FIB to cut a piece of the nanowires, as shown in **new Supplementary Fig. 16**. The FFT images of both the foil and nanowires confirm the single crystal nature of the Cu nanowires inheriting from the starting single-crystal Cu foil.

Supplementary Fig. 16 | TEM and HRTEM images of single crystal Cu nanowire.

a, SEM image of single crystal Cu nanowires, **b**, TEM image of Cu nanowires rooted on Cu foil fabricated by FIB. The cover layer on the nanowires is Pt layer, which is used to protect the nanowires during FIB cutting process. **c**, Topography of the bulk Cu foil, the red arrow in (b) points to the area of (c). The inserted clear HRTEM image shows ordered atoms of Cu foil. **d**, Diffraction pattern of the Cu foil in b, which shows a typical fcc single crystal structure. **e,f**, TEM images at the regions denoted by A and B in (b). The inserted FFT images of (e) and (f) indicated the single crystal nature of the Cu nanowire as well. The inserted HRTEM shows the ordered atoms of Cu nanowires. Scale bars, 1 μm (**a**), 100 nm (**b**), and 10 nm (**c**, **e**, and **f**), 2 nm (inserted HRTEM in **c**, **e**, **f**) and 10 nm^{-1} (**d**).

Our revision #7: We have added the new results in revised Supplementary Information

(Supplementary Figs. 14-16). We have also amended the discussion in the revised Manuscript (highlighted in page 8).

5. The reviewer noted that only imprinting force was given (e.g. page 7) in the experiments, this doesn't make sense if it is not related with the sample size.

Our response:

We thank the reviewer for this important comment. We have added this information to the Method section of the revised Manuscript (highlight in page 15).

Our revision #8:

“Before ultrasonic nanoimprinting, metal foils and AAO molds were cut into square chips with a side length of approximately 5 mm.”

6. Nanoimprinting of pre-deposited multilayer thin film may introduce uniformity issues because of the rough surface of sample, mold, or indenter.

Our response:

We fully agree that the surface of both the mold and metal foil are rough. We are still seeking uniform substrates and molds to improve the performance of our technique.

One way is to use silicon wafer as substrates, which is very smooth [Li, J., Liu, Y., Dai, Y., Yue, D., Lu, X. and Luo, J., 2013. Achievement of a near-perfect smooth silicon surface. *Science China Technological Sciences*, 56(11), pp.2847-2853.]. We attempted the deposition of 200-nm-thick Au film on silicon wafer to fabricate the nanowires, as shown in new **Supplementary Fig. 8**. In this way, the uniformity is much improved.

However, the uniformity of AAO mold needs further improvement. Further alternative could be customized hard mold (such as titanium, tungsten, and alumina) patterned by standard microfabrication process, *e.g.*, EBL, followed by film deposition (*e.g.*, metal film) or etching (*e.g.*, alumina). However, engineering optimization of the customization of these molds requires further investigation.

Supplementary Fig. 8 | SEM image of fabricated Au nanowires on silicon wafer.

Before ultrasonic nanoimprinting process, 200 nm-thick Au layer was deposited on silicon wafer. Scale bar, 1 μm and 100 nm (insert).

Our revision #9: we have added the new results to Supplementary Information (Supplementary Fig. 8).

7. The authors are suggested to give the force constant for the indenter surface in the MD methods.

Our response:

We thank the reviewer for this important comment. In the MD methods, the force constant for all the employed the indenter surface was $1000.0 \text{ eV}/\text{\AA}^3$.

Our revision #10: We have added this information in the revised Manuscript (highlight in page 17).

“A virtual planar indenter (with force constant of $1000 \text{ eV } \text{\AA}^{-3}$) on the top surface of the Ag samples was used...”

8. The authors show the averaged imprinting force (mean force during each loading half-cycle) in the simulations (e.g. Fig. 3j). Can the authors show the time dependent reaction force within one cycle?

Our response: Thank you for the constructive suggestions. As per the reviewer’s suggestions, we have provided the time-force curve of Cycle 6 under the loading pattern $\delta = 0.14$ as follows (corresponding to our Fig. 3).

Reviewer #2 (Remarks to the Author):

Ge et al. report a novel ultrasonic nanoimprinting method that produces nanostructures from all solid materials despite their chemical reactivity. In brief the technique consists in placing an AAO template in contact with the foil to be patterned and apply an ultrasonic horn. The process shows high resolution (replicating the AAO template) with high aspect ratios (up to 100 microns length nanowires). The “nanojackhammer” can also produce nanowires from multilayered foils as demonstrated with some examples. The authors also report the physical mechanisms underlying the technique and fabricate and test a plasmonic structure for the SERS detection of pyocyanin.

In general, this is an interesting novel nanofabrication technique. Its main advantages are that it allows rapid templating with nanoscale features of a given solid at RT and within minutes. However, it also faces some drawbacks such as demolding requires in some cases the dissolution of the template. The authors have shown a variety of nanostructures possible, these nanowires can also be obtained with alternative methods (eBL + RIE), although with additional cost and time. In sum, this technology can be interesting for the community, this is why this manuscript may become acceptable for publication should the following comments be addressed by the authors:

Q1. An interesting advantage of the technique is that it preserves the crystal structure of the original foil, however I do not see that this benefit has been exploited fully by the authors to produce single crystal structures. Could single crystal nanostructures be achieved if used a single crystal as foil, such as a silicon wafer or an epitaxial gold film?

Our response:

We thank the reviewer for your positive comments and constructive suggestions.

Silicon wafers could be challenging because it is thick and very brittle, while thin epitaxial silicon film is possible similar to the thin alumina film in Figure 4. To address the reviewer's question, we imprinted Cu nanowires on single crystal (111) Cu foil. The following XRD spectra (new **Supplementary Fig. 15b**) of the Cu foil shows only one peak [Cu (111)], indicating the single crystal nature of the Cu foil.

Supplementary Fig. 15 | X-ray diffraction spectra of (a) polycrystalline Cu foil and (b) single crystal Cu foil.

After imprinting Cu nanowires from the single crystal Cu foil, we used FIB to cut a piece of the nanowires, as shown in new **Supplementary Fig. 16**. The FFT images of both the foil and nanowires confirm the single crystal nature of the Cu nanowires inheriting from the starting single-crystal Cu foil.

Supplementary Fig. 16 | TEM and HRTEM images of single crystal Cu nanowire.

a, SEM image of single crystal Cu nanowires, **b**, TEM image of Cu nanowires rooted on Cu foil fabricated by FIB. The cover layer on the nanowires is Pt layer, which is used to protect the nanowires during FIB cutting process. **c**, Topography of the bulk Cu foil, the red arrow in (b) points to the area of (c). The inserted clear HRTEM image shows ordered atoms of Cu foil. **d**, Diffraction pattern of the Cu foil in b, which shows a typical fcc single crystal structure. **e,f**, TEM images at the regions denoted by A and B in (b). The inserted FFT images of (e) and (f) indicated the single crystal nature of the Cu nanowire as well. The inserted HRTEM shows the ordered atoms of Cu nanowires. Scale bars, 1 μm (**a**), 100 nm (**b**), and 10 nm (**c**, **e**, and **f**), 2 nm (inserted HRTEM in **c**, **e**, **f**) and 10 nm^{-1} (**d**).

Our revision: We have added the new results in revised Supplementary Information

(Supplementary Figs. 15 and 16).

Q2. What is the interplay between the hardness/mechanical properties of the alumina template and the target material? could other materials work as stamps or this technique works only with AAO as stamp? AAO is limited to hexagonal arrays and the domains are rotated to each other... Is it possible to replicate EBL defined motifs?

Our response:

We thank the reviewer for this interesting and constructive suggestion. For the first question, during the nanoimprinting process, the nanoscale wall of AAO acts as “energy director” to deform the metal foil. Owing to the high-frequency vibration, the energy focuses on the nanowalls of AAO with a reduction of energy loss. Therefore, AAO can be used as a template. Following this rationale, any other nanoscale materials which are hard and stiff (*e.g.*, hard metals Ti and W) can be used also as templates including template patterned by EBL or other techniques.

Q3. Is there conformal contact between AAO molds and metal foil achieved during the printing process? Is it required as in other nanoimprinting methods?

Our response: Nanoimprinting includes hard mold, soft mold, and hybrid mold imprinting techniques. For example, with PDMS template as a soft mold, the low Young’s modulus and low surface energy of PDMS allow conformal contact [Kwon, B., and Jong H. Kim. "Importance of molds for nanoimprint lithography: hard, soft, and hybrid molds." *Journal of Nanoscience* 2016 (2016).]. For hard mold and hybrid mold nanoimprinting, it is challenging to achieve conformal contact.

Our technique belongs to hard mold nanoimprinting, and thus it is difficult to get 100% conformal contact of AAO and foil before nanoimprinting. However, during nanoimprinting, metal foil will be flattened and deformed, and then conformal contact may be achieved during and after nanoimprinting.

Q4. The authors mention a Au-polycarbonate nanowire array. Could the authors elaborate more on the application of their method to organic materials?

Our response:

Plastic nanoimprinting based on ultrasonic nanoimprinting have been reported several years ago, but they used microscale mold as the template, *e.g.*, micro scale Ni mold. [Mekaru, H., Noguchi, T., Goto, H. and Takahashi, M., 2008. *Effect of applying ultrasonic vibration in thermal nanoimprint lithography. Microsystem technologies*, 14(9-11), pp.1325-1333.]. In our work, we use AAO as the template to achieve nanoscale structure of polymer polycarbonate. We have tried poly(methyl methacrylate), polystyrene, and polypropylene. All these materials are soft and have poor heat conductivity. During the nanoimprinting process, the “energy director” with high-frequency vibration will locally heat the AAO and polycarbonate at contact points, softening the polymer to flow into the pores; similar to those in microscale fabrication. One of the unique applications that we are exploring is to make microplastic samples for studying their environmental impacts. Microplastic, defined a decade ago, has huge impact on the health of human, who can unintentionally eat up to half kilogram per year. Thus, our nanoimprinting method could make unique contribution to this research. In general, we can contribute to any applications that need nanoscale organic materials

(*e.g.*, organic nanowires for electronic devices).

Q5. The authors mention in their ref 43 another system to measure QS. How does their structure compare in terms of sensing performance to this reference?

Our response:

As the setup of our Raman imaging system was different from that used in Ref 43, the detection sensitivity cannot be compared fairly. Nevertheless, the signal to noise ratio of the spectra we have obtained is at a similar level to those shown in Figure 6b of Ref 43; each peak of the Raman spectra was clearly resolved in our experiment. Compared with the detection sensitivity, spectrum reproducibility is even more important in quantitative detection of QS. As we have demonstrated in our manuscript, the point-to-point signal variation was below 10%, which ensures reliable measurement. However, no such data was presented in the cited paper (Ref 43) for comparison. Moreover, the signal uniformity across a sample area is excellent (Supplementary Fig. 32). In summary, our measurement is reliable and sensitive, similar to that in ref 43.

Q6. The manuscript is well written and organized. However, in some parts, I would appreciate a bit more experimental detail, i.e.: in page 5, the authors describe the nanoimprinting process of a metal foil, but the metal and how was it obtained are not specified therein. Similarly, on page 8, the nanostructuring process of an Au-Bi multilayer lacks the information of this material. All this info can be found at the experimental section, but a bit more information helps the reader to grasp the idea more rapidly.

Our response: Thank you for your constructive suggestion. As per the reviewer's suggestion, we have added the following description in the revised manuscript (highlighted in page 5 and page 8).

Our revision #11:

1. “e.g., a 100- μm -thick commercial metal foil (Fig. 1a); then...” (page 5)
2. “a nanowire array containing Bi-Au heterojunctions was successfully fabricated by depositing Au and Bi layers on a commercial Au foil” (page 8)

Reviewer #3 (Remarks to the Author):

In this manuscript, the authors propose a scalable and rapid nanostructures-imprinting technique, named nanojackhammer. Using ultrasonic cyclic loading, the authors demonstrate that the imprinting technique works well even at room temperature for several metals (Ag, Au, Cu, Sn, Bi), metal-metal-oxide, and metal-plastic and fabricates length-controlled nanowires having complex nanostructures. Besides, the underlying severe plastic deformation mechanism is discussed based on atomistic behavior in the substrate obtained by molecular dynamics imprinting simulations for FCC metals. This is well-written manuscript with novel technique and demonstration that could become publishable in Nature Communications. However, before that, I suggest the following points need to be addressed to strengthen the paper's discussion.

1. For the cyclic loading frequency, the authors applied an ultrasonic frequency of 20kHz. Is this the best frequency for any imprinting materials, sizes (mold shape), and loading amplitudes? I suggest a frequency-dependent imprinting efficiency can be presented and discussed, which should be valuable information in this technique's practical use. Moreover, since these factors determine strain rate distribution generating in the imprinting material, the information may also strengthen the mechanism discussion based on the MD simulations with a much higher GHz frequency.

Our response:

We greatly appreciate your favorable and encourage comments, as well as constructive

suggestions. We fully agree with the reviewer that frequency is highly related with the performance of nanoimprinting, but frequency of our machine is fixed at 20 kHz. We can only compare 20 kHz with zero frequency (direct loading as shown in Supplementary Figs 3 and 22), and have proved that 20 kHz ultrasonic assisted nanoimprinting indeed lowered the needed energy, achieving metal nanowire fabrication at room temperature. To address reviewer's question, we explain the frequency dependence as follows.

The forming tool (AAO in our work) would concentrate intense ultrasound at the points where the deformation takes place in ultrasonic-assisted deformation. [*Langenecker, B., 1966. Effects of ultrasound on deformation characteristics of metals. IEEE transactions on sonics and ultrasonics, 13(1), pp.1-8; Langenecker, B., Fountain, C.W. and Jones, V.O., 1964. Ultrasonics: an aid to metal forming? Metal Progress, 85, p.97.*]. At these points, the ultrasonic energy causes the speed of deformation to increase under higher applied frequency, because metals can absorb more energy from sound waves when higher frequency was applied, and the shear stress would be reduced accordingly. [*Siddiq, A. and El Sayed, T., 2011. Acoustic softening in metals during ultrasonic assisted deformation via CP-FEM. Materials Letters, 65(2), pp.356-359.*]. Therefore, in principle when the frequency increases, the needed loading force will decrease. Compared with zero frequency (direct loading nanoimprinting in our work), the loading force needed for ultrasonic nanoimprinting is reduced by 100 times. Long Cu nanowires (> 15 μm) can be obtained under 1,200 N in ultrasonic nanoimprinting while only a dent was made on Cu foil under 88,000 N in direct loading nanoimprinting

(Supplementary Fig. 3 and new Supplementary Fig. 13).

2. The authors performed MD simulation on FCC metals and discuss the imprinting mechanism based on dislocation behavior during the cyclic loading, such as alternating dislocation generation and recovery mechanism. As the authors mention in the manuscript, even a significantly higher strain rate was used in MD simulation. I believe that the simulations should demonstrate the essence of the imprinting mechanism of FCC metals and maybe glimpse the fundamental physics of the mean force reduction. However, imprinting not only FCC metals but also oxide and plastic materials, and perhaps also ceramics and glasses, should be one of the advantages of this imprinting technique. Since the non-FCC solid materials generally have different plastic deformation mechanisms in such nano-scaled confined volume, more general discussion, which is applicable even for the non-FCC materials, is needed.

Our response:

We thank the reviewer for this important comment. We fully agree that non-FCC solid materials have different deformation mechanisms due not only to their intrinsic nanostructures and properties, but also to ambient conditions like temperature and pressure. For crystalline metals, FCC or non-FCC metals, the deformation mechanism during nanoimprinting is expected to similarly involve dislocation generation and recovery, although fine details can differ. FCC structures are typical in crystalline metals; therefore, we chose FCC structures to study the metal deformation mechanism in our paper.

As pointed out by the reviewer, unlike crystalline metals with plasticity at room temperature that can experience dislocations nucleation and propagation, materials like oxides, plastics, ceramics, metallic glasses and glasses are generally thermal plastic. The thermal conductivity of these materials is two orders of magnitude lower than that of metals (**Supplementary Table 4**). Therefore, during ultrasonic nanoimprinting, the deforming parts of these materials would be heated, and their thermoplastic forming abilities are typically utilized to induce a softening behavior under imprinting [Kumar, G.; Tang, H. X.; Schroers, J., *Nanomoulding with amorphous metals. Nature* 2009, 457 (7231), 868-872; Guo, L. J., *Nanoimprint Lithography: Methods and Material Requirements. Adv. Mater.* 2007, 19 (4), 495-513; Schroers, J., *Processing of Bulk Metallic Glass. Adv. Mater.* 2010, 22 (14), 1566-1597.]. However, the imposing of thermal effects can bring a new question, *i.e.*, the trade-off between imprinting temperature and thermal stability. Nevertheless, measurement of the local heating temperature is extremely challenging in our current setup as the deformation point caused by AAO is less than 100 nm. Further investigation is thus needed to fully clarify the cooperative roles of the imprinting temperature and ultrasonic loading as well as the intrinsic deformation mechanism in the imprinting of thermal-plastic materials.

Material	Thermal diffusivity (mm²/s)	Thermal conductivity (W/mK)	Melting temperature (°C)
Ag	166	420	961
Au	127	318	1063

Cu	111	390	1083
Al	97	220	660
Plastics	~0.1	~0.1	~200

Supplementary Table 4. Summary of the thermal properties of common metal materials and plastic materials.

Our revision #12: we have added new Supplementary Table 4 with associated discussion in revised Supplementary Information (highlight in page 42).

3. Minor points:

(i) In the MD simulation, the effect of boundary conditions (BC) applied to the substrate (PBC in x and y and maybe almost rigid in z) should be carefully checked. When the imprinting proceeds, if the total number of atoms is constant as in this simulation, the substrate thickness must be reduced, and the BC effects should be changed accordingly.

Our response:

We thank the reviewer for this important comment. Actually, PBCs are imposed in x and y direction. In the z direction, the BC is non-periodic and shrink-wrapping, instead of being rigid, which means that the position of the face is set as to encompass the atoms in z-dimension. It can be seen from our Fig. 3b-e that, as the cycle number increases, the thickness (or height in z direction) over the mold is reduced.

Our revision #13: We have added this information in Method section of the revised Manuscript (highlight in page 17).

(ii) For Fig.3, the definition of volume fraction of dislocation is not clear.

Our response:

Thank you for your careful reading. We have changed the legend from the “volume fraction of dislocations” to “volume fraction of dislocated atoms”, to be precise. The dislocated atoms are defined as atoms that are not of FCC type. Please see the new figure as follows.

Our revision #14: Figure 3 has been revised, and the definition is added into revised

Figure 3 caption.

Revised Fig. 3 | Atomistic simulations of nanoimprinting on silver substrate under direct ($\delta = 1$) and cyclic ($0 < \delta < 1$) loading patterns. a, Load-imprinting displacement curves. Under direct loading ($\delta = 1$), the imprinting load is monitored

every 0.1 nm of the displacement. Under cyclic loading ($0 < \delta < 1$), the imprinting load is defined as the mean force during each loading half-cycle. **b-e**, Sectional views of the deformed silver substrate at imprinting cycles 5 and 6 under $\delta = 0.14$. Atoms are colored according to their local lattice symmetry. **f-i**, Volume fraction of dislocated atoms (HCP and BCC type) along the length (z -axis) of the imprinted nanowire at moments corresponding to **(b-e)**. The dashed lines in **(f-i)** and in **(b-e)** indicate the position $z=0$. **j**, the steady-state mean imprinting force as a function of the loading parameter.

REVIEWERS' COMMENTS

Reviewer #1 (Remarks to the Author):

Since the authors have addressed all the comments of reviewers, I suggested to publish after minor revision.

1. Please clarify the inconsistent in your responses: in the response to my concerns 1, the authors claimed “For the imprinted metal sample, in our scenario with the imprinting frequency of ~20 kHz, the absorbed energy from sound waves should be negligible [Langenecker, B., Effects of Ultrasound on Deformation Characteristics of Metals. IEEE Transactions on Sonics and Ultrasonics 1966, 13 (1), 1-8.]” and they neglected the absorbed energy when estimating the temperature rise due to plastic deformation. While in the response to the point 1 of Reviewer #3, the authors claimed “At these points, the ultrasonic energy causes the speed of deformation to increase under higher applied frequency, because metals can absorb more energy from sound waves when higher frequency was applied, and the shear stress would be reduced accordingly. [Siddiq, A. and El Sayed, T., 2011. Acoustic softening in metals during ultrasonic assisted deformation via CP-FEM. Materials Letters, 65(2), pp.356-359.]”

2. The authors should give the experimental details of nanoimprinting of Ni (Supplementary Fig. 12).

Reviewer #2 (Remarks to the Author):

The authors have included plenty of additional new experiments and discussions that improve the clarity of their work and reply to the reviewers. I am happy to recommend this revised version for publication in Nature comms.

Reviewer #3 (Remarks to the Author):

I believe that the authors have addressed all the comments and suggestions that all the reviewers asked. Now the paper has excellent shape and so provides a great contribution to the nano-manufacturing field. It should be published in Nature Communications.

Point-by-Point Response to the Reviewers' Comments (NCOMMS-20-34796A)

We thank all the reviewers for their insightful comments and constructive suggestions, which have allowed us to further improve our manuscript. Our point-by-point responses are as follows.

REVIEWER COMMENTS

Reviewer #1 (Remarks to the Author):

Since the authors have addressed all the comments of reviewers, I suggested to publish after minor revision.

1. Please clarify the inconsistent in your responses: in the response to my concerns 1, the authors claimed “For the imprinted metal sample, in our scenario with the imprinting frequency of ~20 kHz, the absorbed energy from sound waves should be negligible [Langenecker, B., Effects of Ultrasound on Deformation Characteristics of Metals. IEEE Transactions on Sonics and Ultrasonics 1966, 13 (1), 1-8.]” and they neglected the absorbed energy when estimating the temperature rise due to plastic deformation. While in the response to the point 1 of Reviewer #3, the authors claimed “At these points, the ultrasonic energy causes the speed of deformation to increase under higher applied frequency, because metals can absorb more energy from sound waves when higher frequency was applied, and the shear stress would be reduced

accordingly. [Siddiq, A. and El Sayed, T., 2011. Acoustic softening in metals during ultrasonic assisted deformation via CP-FEM. Materials Letters, 65(2), pp.356-359.]”

Our response:

We thank the reviewer for this important comment. In response to the reviewer’s concern 1, we discussed about sound wave’s contribution to thermal transport. During ultrasonic nanoimprint, the associated heat release mainly comes from local plastic energy dissipation at the corner, direct sound waves-induced heat is negligible (compared to local plastic energy-induced heat) based on this reference [Langenecker, B., Effects of Ultrasound on Deformation Characteristics of Metals. IEEE Transactions on Sonics and Ultrasonics 1966, 13 (1), 1-8.]

In response to point 1 of Reviewer #3, we explained the working principle of nanoimprinting - “R.T nanoimprinting is essentially a severe local plastic deformation process”, where sound wave energy is absorbed by metal and then causes severe plastic deformation [Siddiq, A. and El Sayed, T., 2011. Acoustic softening in metals during ultrasonic assisted deformation via CP-FEM. Materials Letters, 65(2), pp.356-359].

In other words, energy from sound wave is mainly utilized for generating severe plastic deformation instead of directly inducing heat. Thus, there is no conflict between these two responses.

2. The authors should give the experimental details of nanoimprinting of Ni (Supplementary Fig. 12).

Our response:

We thank the reviewer for the constructive suggestion. To address reviewer's question, we have added following description in the revised manuscript:

“100- μm -thick commercial metal foils (Au, Ag, Cu, Sn, Al, Bi, and Ni; purity of 99.999%) were purchased from JIARUN Metal Materials Company.” (page 15, materials and characterization section).

The nanoimprinting process of Ni nanowires is described on page 15 (ultrasonic of polycrystalline metal nanowires/nanorods section) as “Before ultrasonic nanoimprinting, metal foils and AAO molds were cut into square chips with a side length of approximately 5 mm. After ultrasonic nanoimprinting, the AAO molds/substrate stack were immersed in 1 M NaOH solution for up to 2 h to remove AAO and then expose the nanowires...”. Also, the summary of nanoimprinting parameters used to obtain Ni nanowires is shown in supplementary table 2, “loading force = 1,200 N, amplitude = 25%, and holding time = 30 s”.

Reviewer #2 (Remarks to the Author):

The authors have included plenty of additional new experiments and discussions that improve the clarity of their work and reply to the reviewers. I am happy to recommend this revised version for publication in Nature comms.

Our response:

Thank you very much for all your time and effort in reviewing our manuscript. Your valuable comments and constructive suggestions have significantly improved the

quality of our manuscript.

Reviewer #3 (Remarks to the Author):

I believe that the authors have addressed all the comments and suggestions that all the reviewers asked. Now the paper has excellent shape and so provides a great contribution to the nano-manufacturing field. It should be published in Nature Communications.

Our response:

Thank you very much for all your time and effort in reviewing our manuscript. Your valuable comments and constructive suggestions have significantly improved the quality of our manuscript.